# HSRL-2 aerosol optical measurements and microphysical retrievals vs. airborne in situ measurements during DISCOVER-AQ 2013: an intercomparison study

Patricia Sawamura<sup>1,2</sup>, Richard H. Moore<sup>1</sup>, Sharon P. Burton<sup>1</sup>, Eduard Chemyakin<sup>1,3</sup>, Detlef Müller<sup>4,3</sup>, Alexei Kolgotin<sup>5</sup>, Richard A. Ferrare<sup>1</sup>, Chris A. Hostetler<sup>1</sup>, Luke D. Ziemba<sup>1</sup>, Andreas J. Beyersdorf<sup>1</sup>, and Bruce E. Anderson<sup>1</sup>

<sup>1</sup>NASA Langley Research Center, Hampton, VA, USA
 <sup>2</sup>Universities Space Research Association, Columbia, MD, USA
 <sup>3</sup>Science Systems and Applications, Inc., Hampton, VA, USA
 <sup>4</sup>University of Hertfordshire, Hatfield, Hertfordshire, UK
 <sup>5</sup>Physics Instrumentation Center, Troitsk, Russia

Correspondence to: P. Sawamura (patricia.sawamura@nasa.gov)

**Abstract.** Over 700 vertically-resolved retrievals of effective radii, number, volume, and surface-area concentrations of aerosols obtained from inversion of airborne multiwavelength High Spectral Resolution Lidar (HSRL-2) measurements are compared to vertically resolved airborne in situ measurements obtained during DISCOVER-AQ campaign from 2013 in California and Texas. In situ measurements of dry and humidified scattering, dry absorption, and dry size distributions are used to estimate

- hygroscopic adjustments which, in turn, are applied to the dry in situ measurements before comparison to HSRL-2 measurements and retrievals. The HSRL-2 retrievals of size parameters agree well with the in situ measurements once the hygroscopic adjustments are applied to the latter, with biases smaller than 25% for surface-area concentrations, and smaller than 10% for volume concentration. A closure study is performed by comparing the extinction and backscatter measured with the HSRL-2 with those calculated from the in situ size distributions and Mie theory, once refractive indices (at ambient RH) and hygro-
- scopic adjustments are calculated and applied. The results of this closure study revealed discrepancies between the HSRL-2 optical measurements and those calculated from in situ measurements, in both California and Texas datasets, with the aerosol extinction and backscatter coefficients measured with the HSRL-2 being larger than those calculated from the adjusted in situ measurements and Mie theory. These discrepancies are further investigated and discussed in light of the many challenges often present in closure studies between in situ and remote sensing systems, such as: limitations in covering the same size range of
- particles with in situ and remote sensing instruments, as well as simplified parameterizations and assumptions used when dry in situ data are adjusted to account for aerosol hygroscopicity.

# 1 Introduction

In situ and remote sensing airborne measurements are of fundamental importance for the study of aerosol-cloud-climate interactions (McFarquhar et al., 2011; Baumgardner et al., 2011). Ground- and space-based observations are just as important,

but are not enough for providing a comprehensive observational basis to evaluate and improve air quality and climate models. Ground-based observations are very helpful for studying the temporal evolution of aerosol and clouds with higher temporal resolution than satellite measurements can provide, but they lack in spatial coverage. Satellites, on the other hand, provide measurements over a wider spatial range but suffer from the low temporal resolution of measurements obtained at any particular location on the globe. Aircraft measurements offer a solution to bridge this gap between ground- and space-based observations,

5

and are also important for testing the feasibility of future space-borne instruments.

However, as McFarquhar et al. (2011) point out, more effort is needed to improve our current understanding of the caveats and uncertainties associated with airborne measurements. Kassianov et al. (2015) emphasize the importance of closure studies to ensure consistency among the many different measurements that can be obtained from an airborne platform.

- 10 Comprehensive observations of aerosol optical and microphysical properties are also critical for developing and evaluating aerosol transport model parameterizations and assessing global aerosol-radiation impacts on climate. Lidars, in particular, are important tools for atmospheric measurements as they are able to provide detailed information on the vertical distribution of aerosols.
- NASA Langley Research Center's HSRL-2 (High Spectral Resolution Lidar-2) is a unique airborne system that provides a 15 complete  $3\beta + 2\alpha$  data set (i.e. 3 backscatter + 2 extinction) that allows the retrieval of horizontally- and vertically-resolved aerosol physical properties, such as number, surface-area, and volume concentrations, as well as effective radius, in addition to the multiwavelength optical properties (Müller et al., 2014). HSRL-2 is also a prototype for the spaceborne lidar system being considered for the ACE (Aerosol-Cloud-Ecosystem) mission (http://dsm/gsfc.nasa.gov/ace) which has the objective of improving our understandings of the interactions among aerosols, cloud and precipitation systems, and ocean ecosystems.
- A number of studies have focused in the  $3\beta + 2\alpha$  retrieval technique in the past decade (Müller et al., 1998, 2001; Wandinger et al., 2002; Müller et al., 2003; Murayama et al., 2004; Müller et al., 2006; Tesche et al., 2008; Noh et al., 2009; Balis et al., 2010; Alados-Arboledas et al., 2011; Navas-Guzmán et al., 2013; Nicolae et al., 2013; Sawamura et al., 2014), most of which used data from ground-based Raman lidars. The validation of such retrievals, however, has always been a challenging task due to the unavailability of direct colocated measurements.
- 25 Direct measurements of aerosol size distributions, for instance, can only be obtained with in situ instruments. Most in situ measurements, however, are obtained at ground level, limiting the comparison to the few lidar retrievals that are obtained closest to the ground.

Other studies have compared the  $3\beta + 2\alpha$  lidar retrievals to AERONET (Aerosol Robotic Network, (Holben et al., 1998)) retrievals (Veselovskii et al., 2009; Sawamura et al., 2014). However, AERONET total column retrievals of volume concen-

30 trations are not directly comparable to vertically-resolved aerosol measurements, requiring an assumption that the aerosol is uniformly mixed throughout the boundary layer. Intercomparison studies between  $3\beta + 2\alpha$  lidar retrievals and AERONET retrievals or ground-based in situ measurements, therefore, are not suitable to properly evaluate the performance of the lidar microphysical retrievals obtained for different altitudes.

During DISCOVER-AQ, HSRL-2 and a suite of in situ instruments were deployed onboard two aircraft. The aircraft carrying the in situ instruments flew in spirals, providing vertical profiles of optical, physical, and chemical properties of aerosols. In this

5

study we present the intercomparison of aerosol microphysical retrievals obtained with the HSRL-2 data to vertically-resolved in situ measurements obtained during two deployments of DISCOVER-AQ.

The hygroscopicity of aerosols represents a challenge for comparison studies like this, i.e. lidar vs. in situ (Zieger et al., 2010, 2011; Sawamura et al., 2014). At high relative humidities aerosol particles experience changes in size which in turn translates into changes in their optical properties. These changes depend on the aerosol chemical composition and also on the ambient relative humidity (RH). Remote sensing measurements, like those from HSRL-2, are obtained under ambient conditions. In situ measurements, on the other hand, are usually obtained under dry conditions (low relative humidity). Therefore, in order to avoid an unrepresentative comparison to lidar measurements and/or retrievals, in situ measurements have to be adjusted to account for hygroscopic effects.

- Müller et al. (2014) presented the first results of microphysical retrievals from HSRL-2 data obtained during the Two-Column Aerosol Project (TCAP) (Berg et al., 2016). During TCAP, the aircraft carrying the in situ instruments flew in spirals on 2 days, allowing the comparison of a full profile (up to  $\sim 4$  km) of colocated in situ measurements and lidar retrievals of number, surface-area, and volume concentration, as well as effective radii. For this specific comparison case, hygroscopic adjustments were not necessary. Good agreement was observed between the HSRL-2 retrievals and the in situ measurements.
- In this study we present a methodology to adjust the in situ measurements to account for aerosol hygroscopicity and discuss the results from the comparison of more than 700 lidar retrievals of size parameters, i.e. number, surface-area, and volume concentrations, and effective radius, to colocated in situ measurements obtained during DISCOVER-AQ 2013. We analyze approximately 110 colocated profiles of HSRL-2 retrievals and in situ measurements (700+ sets of  $3\beta + 2\alpha$ ), making this the largest study to date on the evaluation of microphysical retrievals obtained from multiwavelength lidar measurements.
- In a closure study, backscatter and extinction coefficients were calculated using the in situ size distributions already adjusted to ambient RH, the corresponding adjusted refractive indices, and Mie theory. These calculated optical data set are then compared to the measurements obtained by the HSRL-2 and the results are analyzed and discussed in light of the many challenges that are often present in studies of this kind, such as: limitations in covering the same size range of particles with in situ and remote sensing instruments, as well as simplified parameterizations and assumptions used when dry in situ data are adjusted to account for aerosol hygroscopicity.

# 2 DISCOVER-AQ

DISCOVER-AQ stands for <u>Deriving Information on Surface Conditions from COlumn and VER</u>tically Resolved Observations Relevant to <u>Air Quality</u>. For brevity we will refer to DISCOVER-AQ as DAQ for the remainder of this paper. DAQ was a 4-year NASA project with the objective of improving the interpretation of total-column satellite observations to help diagnose

30 near-surface conditions relating to air quality. DAQ took place in Baltimore-Washington D.C. in the Summer 2011, in San Joaquin Valley, California and Houston, Texas in Winter and Summer 2013, respectively, and in the Colorado Front Range in Summer 2014.

In this study we use data from the 2013 deployments in California (DAQ CA: Jan/Feb 2013) and Texas (DAQ TX: Aug/Sep 2013).

During DAQ 2013, HSRL-2 was flown onboard the NASA Langley B200 King Air aircraft downward-looking at approximately 8.5 km altitude. Many in situ instruments were flown onboard NASA Wallops P-3B aircraft which spiraled up and down over several designated ground stations from altitudes as low as 15 m up to about 5 km. The two aircraft followed coordinated flight tracks, both flying over the designated air quality monitoring ground stations with close time coincidence, allowing for colocation of measurements from the in situ instruments suite and the HSRL-2. Figure 1 shows maps of the King Air and P-3B

aircraft flight tracks during DAQ California and Texas, as well as the designated ground stations.

# 3 LaRC HSRL-2

HSRL-2 follows in the heritage of the successful airborne instrument HSRL-1 (Hair et al., 2008) which independently measures the aerosol extinction and backscatter at 532 nm with the HSRL technique (Shipley et al., 1983), and the backscatter at 1064 nm with the standard backscatter lidar technique (Fernald, 1984). HSRL-1 also measures aerosol depolarization ratio at 532 nm and 1064 nm.

HSRL-2 provides independent measurements of aerosol extinction and backscatter at 355 nm using the HSRL technique, and depolarization ratio at 355 nm, in addition to the measurements that can be obtained with HSRL-1. These new capabilities make HSRL-2 the first airborne system that provides a complete  $3\beta + 2\alpha$  dataset (i.e. 3 backscatter + 2 extinction) from which microphysical properties of aerosols can be retrieved (Qing et al., 1989; Müller et al., 1999a).

HSRL-2 data are sampled at temporal resolution of 0.5 second and vertical resolution of 15 meters. Aerosol backscatter and depolarization products are horizontally averaged for 10 seconds ( $\sim 1$  km at nominal aircraft speed) and aerosol extinction

products are averaged for 60 seconds (~ 6 km). The microphysical data products were retrieved at vertical resolution of 150 m during DISCOVER-AQ Texas and 75 m during DISCOVER-AQ California due to shallow planetary boundary layers observed throughout the campaign.

## 3.1 Inversion algorithm

- The inversion method used for the retrievals is based on the concepts described by Müller et al. (1999a) and Veselovskii et al. (2002) and uses the backscatter measurements at 355, 532 and 1064 nm and extinction measurements at 355 and 532 nm. The original inversion method has been successfully applied to a number of case studies by many lidar groups in the last decades (Müller et al., 1998, 2001; Wandinger et al., 2002; Müller et al., 2003; Murayama et al., 2004; Müller et al., 2006; Tesche et al., 2008; Noh et al., 2009; Balis et al., 2010; Alados-Arboledas et al., 2011; Navas-Guzmán et al., 2013; Nicolae et al., 2013; Sawamura et al., 2014).
- The inversion method used with the HSRL-2 data, however, has been modified to allow automated and unsupervised processing of large volumes of data. The algorithm is described by Müller et al. (2014) which we briefly review here.

5

25

The equations that relate the measured aerosol optical properties (i.e.  $3\beta + 2\alpha$ ) to the aerosol's inherent microphysical properties, such as size distribution and complex refractive index, are solved under the assumption that the size distributions can be reconstructed as linear combinations of eight logarithmically equidistant triangular-shaped base functions within an inversion window (Müller et al., 1999a, b). We use a comparably wide range of parameters in the inversion windows, i.e. particle radius from 0.03 - 8 mum, real part of the complex refractive index from 1.325 - 1.8, and imaginary part from 0 - 0.1. In order to stabilize the retrieval results we therefore apply extra constraints to the solution spaces. For that purpose we use the inversion results we have for particle effective radius and number concentration. We compute the mean values of effective radius and number concentration, respectively. The threshold value with regard to deviation

10 of individual solutions from mean values was set to 25%-deviation for effective radius and 100 % for number concentration. The discrepancy averaging interval was set to less than 10 %. The computations are parallelized to increase the speed of the data inversion process.

For each  $3\beta + 2\alpha$  set the algorithm is run 9 times. In 8 of those runs, the extinction and backscatter coefficients input are distorted by their respective uncertainties in different combinations in order to simulate possible measurement error scenarios

15 (see Table 1). Another run is performed with error-free input data. Hundreds of thousands of solutions are obtained with those 9 runs. The 500 solutions with the lowest discrepancies are averaged and stored as the final solution (Müller et al., 1999b; Veselovskii et al., 2002). The standard deviation of those 500 best solutions is assumed to be a good measure of the retrieval uncertainty.

Only 3β+2α sets with uncertainties below 20% are used for the inversion (Müller et al., 1999b). The uncertainties originate
from the HSRL-2 system's random noise and are estimated using the noise scale factor methodology described by Liu et al. (2006).

## 4 In situ instruments

The aerosol size distribution and optical properties are measured in situ by the P-3B aircraft by the NASA Langley Aerosol Research Group (LARGE). Aerosols are sampled isokinetically via a low-turbulence inlet mounted on the port side of the aircraft that is able to transmit particles smaller than 5  $\mu m$  diameter (50% cutoff efficiency) (McNaughton et al., 2007). The

sampling stream is then brought into the cabin and split between multiple instruments, only a subset of which are briefly described here.

Aerosol dry and humidified light scattering coefficients at 450, 550, and 700 nm wavelength are measured by two integrating nephelometers (Model 3563, TSI, Inc., Shoreview, MN, USA) operated in tandem (Pilat and Charlson, 1966; Clarke et al.,

30 2002; Ziemba et al., 2013). The measurements are corrected for truncation errors following Anderson and Ogren (1998). One

of the nephelometers is operated at dry relative humidity (RH < 40%), while the other is operated at elevated relative humidity (RH = 80-85% during DAQ). These measurements are used to derive the aerosol hygroscopicity parameter,  $\gamma$ , as

$$\gamma = \frac{\ln\left(\frac{\sigma_{\rm scat,wet}}{\sigma_{\rm scat,dry}}\right)}{\ln\left(\frac{100 - \rm RH_{dry}}{100 - \rm RH_{wet}}\right)} \tag{1}$$

where  $\sigma_{scat,dry}$  and  $\sigma_{scat,wet}$  are the aerosol light scattering coefficients under dry (RH<sub>dry</sub>) and elevated RH conditions 5 (RH<sub>wet</sub>), respectively.

The ambient RH outside the aircraft is computed using the aircraft static temperature measurement and water vapor concentration measured by an open-path diode laser hygrometer (Diskin et al., 2002), and the ambient aerosol scattering coefficient  $\sigma_{scat,amb}$  at this RH<sub>amb</sub> is then determined as

$$\sigma_{\rm scat,amb}(\rm RH_{amb}) = \sigma_{\rm scat,dry} \left[ \frac{100 - \rm RH_{dry}}{100 - \rm RH_{amb}} \right]^{\gamma}$$
(2)

- 10 This transformation is important for comparing the in situ light scattering measurements to those from the HSRL-2 because sampling aerosols in situ using an isokinetic inlet and bringing the sample stream into the warm cabin inherently changes the particle temperature and, hence the relative humidity, while the HSRL measures the aerosol scattering and extinction under unperturbed, ambient conditions.
- The accuracy of the  $\gamma$  parameter fitting method (Kasten, 1969) has been questioned in the past and other multi-parameter 15 methods have been suggested (Kotchenruther and Hobbs, 1998; Kotchenruther et al., 1999; Carrico et al., 2003; Brock et al., 2016). One of the limitations of the  $\gamma$  parameterization is that, for instance, it cannot accurately describe the hygroscopic scattering enhancement (f(RH)) of particles that present abrupt phase transitions, like pure ammonium sulfate particles.

Dry aerosol absorption coefficients are measured at 450, 532, and 700 nm wavelength using a Particle Soot Absorption Photometer (PSAP; Radiance Research, Shoreline, WA, USA) that was heated to 30°C to prevent water condensation on the filter

substrate. PSAP measurements are corrected for filter scattering following (Virkkula, 2010). Aerosol extinction coefficients are computed as the sum of the scattering and absorption coefficients, neglecting hydration effects on the absorption coefficient, which are highly uncertain and likely to be minimal for the largely non-absorbing aerosols observed during most of DAQ. The scattering Ångström exponent is used to adjust the 550 nm scattering to 532 nm prior to the extinction coefficient calculation. The aerosol dry size distribution is measured using two different optical particle counters: an Ultra-High Sensitivity Aerosol

Spectrometer (UHSAS; Droplet Measurement Technologies, Inc., Boulder, CO, USA) that measures particles with diameters that range from 0.06  $\mu m$  to 1  $\mu m$ , and a Laser Aerosol Spectrometer (LAS Model 3340; TSI, Inc.) that measures particles from 0.09  $\mu m$  to 7.5  $\mu m$  but it is limited by the cutoff size of the aircraft inlet, which in this case allows particles only up to 5  $\mu m$ .

These instruments measure the intensity of light scattered from aerosol particles that pass through a focused laser beam. This intensity is proportional to the particle size and can be predicted with Mie theory (Mie, 1908) if the refractive index of

30 the particles and the optical geometry of the instrument are known. Both instruments are field calibrated with NIST-traceable

5

25

polystyrene latex spheres (ThermoScientific, Inc.). The refractive index of such particles ( $m_{PSL} \sim 1.588$ ), however, is higher than that of ambient aerosol particles.

The effect of the refractive index on optical counter measurements have been investigated and it was found that optical counters often undersize ambient particles. (Liu and Daum, 2000; Ames et al., 2000). Kassianov et al. (2015) demonstrate that ignoring the refractive index correction or using non-representative values can cause up to 40% in bias for the calculated scattering coefficient. For this reason, both instruments are also calibrated with monodisperse ammonium sulfate (AS) aerosols ( $m_{AS} \sim 1.53$ ) classified with a Differential Mobility Analyzer (DMA 3081, TSI, Inc.). For this study, we use the AS size calibration.

## 5 Methodology

10 Most of the in situ data used in this study were obtained at dry conditions (RH< 40%), except for the scattering at 550 nm which was also measured at wet conditions (RH~ 80%). Therefore, in order to properly compare the in situ measurements to the HSRL-2 measurements (and retrievals), it was necessary to adjust the dry in situ measurements to account for hygroscopic effects at the ambient RH.</p>

Most ambient aerosol particles experience hygroscopic growth as the ambient RH increases. As the particle grows, the 15 complex refractive index of the particle also changes.

The growth of a particle due to water uptake is described by the hygroscopic growth factor,  $g(RH, D_{dry})$ , which is defined as the ratio of the diameter of the particle at a certain RH,  $D_{amb}$ , to its dry diameter,  $D_{dry}$ :

$$g(\mathrm{RH}, \mathrm{D}_{\mathrm{dry}}) = \frac{D_{\mathrm{amb}}(\mathrm{RH})}{D_{\mathrm{dry}}}$$
(3)

As the particle grows due to water uptake,  $m_{amb}$  decreases and can be calculated by volume weighting the dry particle's 20 refractive index  $m_{dry}$  with the refractive index of water ( $m_{H_2O} = 1.333 \pm 0i$ , Hale and Querry (1973)) following:

$$m_{\rm amb} = \frac{m_{\rm dry} + m_{\rm H_2O}(g^3 - 1)}{g^3} \tag{4}$$

The volume weighted mixing rule assumes the aerosols are homogeneously mixed as they undergo humidification.

For spherical particles, optical properties like scattering and absorption coefficients can be calculated with Mie theory if the size distribution and the complex refractive index (m) of the aerosol particles are known. In this study we have in situ measurements of scattering and absorption coefficients and size distributions at dry conditions, but the  $m_{dry}$  values are unknown.

Once  $m_{dry}$  is determined, it is possible to use it with its respective size distribution in a hygroscopic growth model to reproduce the scattering coefficient measured at ambient conditions. In this process we are able to infer the effective hygroscopic growth factor,  $\bar{g}$ .

While  $g(\text{RH}, D_{\text{dry}})$  can be measured with a tandem differential mobility analyzer (TDMA, Rader and McMurry (1986)), 30 that instrument was not available during DAQ 2013. Therefore, in this study, the hygroscopic growth factor is assumed to be

5

10

30

diameter-independent. For this reason we refer to it as an effective hygroscopic growth factor  $\bar{g}$ . Under this assumption the entire size distribution shifts to larger diameters by a factor of  $\bar{g}$  when the hygroscopic correction is applied.

This methodology was developed based on Zieger et al. (2010). Zieger et al. (2010) also use the concept of *effective growth factor*, although in their study it is called *apparent growth factor*. In their study a fixed pair of complex refractive index is assumed for all their measurements instead of being determined.

In the following subsections we describe how profiles of  $m_{dry}$ ,  $m_{amb}$  and  $\bar{g}$  are retrieved from the in situ measurements.

## 5.1 Data selection

In order to directly compare the vertical profiles of HSRL-2 measurements and retrievals to in situ measurements, only the data obtained within a 10 km radius from the spiral center and 30 minutes from each other were considered. The spirals diameters were approximately 6 km in California and 8-10 km in Texas.

The vertical profiles of in situ measurements were obtained while the P-3B aircraft profiled the atmosphere in a series of ascending/descending spirals. Therefore, the vertical resolution of the in situ measurements varied with the ascent/descent rate. On average, the vertical resolution for the in situ profiles was about 5 m. For this study, in order to standardize the vertical resolution of the in situ measurements, all profiles were interpolated to match the HSRL-2 15 m vertical resolution using a

15 smoothing spline. For the comparison with the HSRL-2 microphysical retrievals (Section 6), the in situ measurements were further reduced by applying a moving average and undersampling the in situ profiles in order to match the retrievals vertical resolution, i.e. 75 m for DAQ CA and 150 m for DAQ TX.

The HSRL-2 microphysical retrieval algorithm (Müller et al., 2014) is based on Mie theory (Mie, 1908) which describes the scattering of light by spherical particles. Therefore, the HSRL-2 dataset was further screened for data corresponding to high

20 aerosol depolarization ratio at 532 nm ( $\delta_{532}$ ) to avoid signals originated from non-spherical particles. Only measurements with  $\delta_{532} < 5\%$  were considered.

Most of the data used in this study are publicly available in the DAQ website: http://www-air.larc.nasa.gov/missions/ discover-aq/discover-aq.html or using data doi: 10.5067/Aircraft/DISCOVER-AQ/Aerosol-TraceGas.

# 5.2 Part I: Retrieving dry complex refractive index (m<sub>dry</sub>) from optical in situ measurements

In the upper part of Figure 2 a block diagram summarizes the iterative algorithm used to retrieve  $m_{dry}$  using the dry in situ data. The blue terms highlight the portions of the algorithm that use the in situ measurements. The profiles shown at the bottom of Figure 2 exemplify one of the many spirals obtained during DAQ TX. This particular spiral was performed over Channel View, in the Houston area, on September 11<sup>th</sup>, 2013 between 21:06 - 21:15 UTC.

The calculation of scattering and absorption coefficients with Mie theory requires a size distribution and complex refractive index. Here, we use the dry scattering coefficients at 450 nm, 550 nm, and 700 nm measured with the dry nephelometers, the dry absorption coefficients at 532 nm measured with the PSAP, and dry size distributions measured with the UHSAS in order to

retrieve the vertical profiles of complex refractive index (at dry conditions,  $m_{dry}$ ). *m* is assumed to be wavelength-independent.

5

15

30

For a given measured size distribution (dry), the scattering and absorption coefficients are calculated for a grid of real and imaginary parts of the refractive index,  $m_R$  and  $m_i$ , respectively. In this grid,  $m_R$  varies from 1.33 to 1.7 in 0.02 increments and  $m_i$  varies from 0 to 0.03 in 0.001 increments. Assuming a wavelength-independent refractive index, the algorithm searches for the combination(s) of  $m_{dry,R}$  and  $m_{dry,i}$  that, along with the measured size distribution in a Mie code (Bohren and Huffman, 1983), allows the reproduction of the scattering coefficient measured in situ (dry) within 20% and the absorption coefficient

within 2 Mm<sup>-1</sup>. The final pairs of  $m_{dry,R}$  and  $m_{dry,i}$  are obtained by averaging all solutions that met the accuracy constraint. In the bottom part of Figure 2 the black profiles are the measurements obtained with the nephelometer and the PSAP onboard the P-3B. The red profiles were calculated using the UHSAS size distribution and  $m_{dry}$  (see Figure 3). The good agreement between the measured and calculated profiles of scattering and absorption in this figure originates from the strict constraint

10 imposed by the algorithm, but it also demonstrates the consistency among the scattering, absorption, and size distribution measurements obtained with the various in situ instruments (Ziemba et al., 2013).

# 5.3 Part II: Retrieving effective growth factors $(\bar{g})$ and complex refractive index corrected for ambient RH $(m_{amb})$

In this step the algorithm iterates over  $\bar{g}$  values that range from 1 to 2 in 0.01 increments to correct the dry size distributions and the dry refractive index (as in Equations 3 and 4) that are used in a Mie code to calculate  $\sigma_{scat,amb}^{550nm}$ . The final  $\bar{g}$  value is the one that allows the recalculation of  $\sigma_{scat,amb}^{550nm}$  within 1% of the measured value.

Figure 3 shows the block diagram depicting the algorithm just described. The profiles at the bottom of this figure show the comparison between the measured scattering coefficient at 550 nm, corrected for the ambient relative humidity as described by Ziemba et al. (2013), and the reconstructed profile calculated with this methodology. It also shows the profiles of the retrieved  $\bar{g}$ ,  $m_{dry}$  (retrieved in Part I), and  $m_{amb}$ .

20 More discussion on  $\bar{g}$  can be found in Appendix A1.

# 5.4 Part III: Closure study: Optical properties evaluation

As described in Section 4, ambient extinction coefficients at 532 nm ( $\alpha_{532}$ ) were computed from independent in situ measurements of scattering and absorption. Once the in situ size distributions are adjusted to account for hygroscopic effects, they can be used with  $m_{amb}$  in a Mie code in order to compute optical properties, such as extinction and backscatter coefficients.

The comparison between the measured (in situ) and the calculated in situ ambient extinction coefficients at 532 nm showed excellent agreement with biases smaller than 1% for both DAQ CA and TX datasets (not shown), which should be expected given the strict constraint imposed for the retrieval of  $\bar{g}$ .

In addition to  $\alpha_{532}$ , the extinction coefficient at 355 nm ( $\alpha_{355}$ ), and the backscatter coefficients at 355, 532, and 1064 nm ( $\beta_{355}$ ,  $\beta_{532}$ , and  $\beta_{1064}$ , respectively) were also computed with the adjusted in situ measurements. This  $3\beta + 2\alpha$  set calculated with the adjusted in situ measurements was then compared to the  $3\beta + 2\alpha$  set measured with the HSRL-2.

## 6 Results

#### Microphysical properties: Effective radius, and number, surface-area, and volume concentrations for the fine 6.1 mode

A total of 172 sets of coincident profiles (i.e. in situ and HSRL-2) were considered for the analysis of the microphysical properties: 95 from DAQ TX and 77 from DAQ CA. Out of those, 108 profiles had valid data points for both HSRL-2 retrievals 5 and adjusted in situ measurements: 76 from DAQ TX (630 data points) and 32 from DAQ CA (126 data points).

During DAQ CA the shallow boundary layers, commonly observed during the wintertime in the San Joaquin Valley, limited the number of measurements available for comparison. As previously mentioned, the HSRL-2 data were screened to ensure that the data used in the analyses corresponded to low depolarization ratio cases (i.e.  $\delta_{532}