# Peer review of "HSRL-2 aerosol optical measurements and microphysical retrievals vs. airborne in situ measurements during DISCOVER-AQ 2013: an intercomparison study"

_Atmospheric Chemistry and Physics, 2016_

## Referee Comment (RC1) · Anonymous Referee #1 · 12 Jun 2016

Sawamura et al. present in their manuscript a comparison study of airborne lidar and in-situ measurements and retrievals. The measurements were recorded during two aircraft campaigns in 2013 in California and Texas. The in-situ recordings of particle size and light scattering had to be transformed to ambient conditions. For this, the hygroscopic growth factor was retrieved by inverting measurements of a humidified and dry nephelometer, a PSAP and two instruments that measured the particle size distribution. Mie calculations were performed in order to compare the in situ measured backscatter and extinction coefficients to the HSRL-2 retrievals. A clear correspondence between in-situ and lidar retrievals was found. However, distinct discrepancies

were found which are discussed. Additional sensitivity studies were performed to investigate the influence of the limited size ranges of the different instruments and the ifnluence of the used parameterizations.

The manuscript structure and presentation quality have to be substantially improved. The focus is often lost and the reader gets confused by unnecessary technical details and repetitions throughout the manuscript. All technical, retrieval related or campaign specific information should be moved to the methods part. In this way the results can focus on the actual findings. Currently, many isolated and singular sentences make up own paragraphs which makes the manuscript difficult to read. Thematically related topics can often be combined to own paragraphs. The amount of tables and figures should be limited to the important ones (additional information should be moved to the supplementary material). Heading titles and section labelling should be improved as well (e.g. avoiding single subsections without subsequent follower and unnecessary long titles). The conclusions should be more concise and should focus on the lessons learned. Substantial editorial work is therefore unavoidable.

The topic and findings are of interest to the scientific community. However, there are many clarifications and questions to the analysis and its interpretation which have to be adequately answered before publication (see detailed comments below). It is for these reasons that I recommend major revisions.

**Detailed comments**

Comments are given in arbitrary order.

1. Page 1, line 10-13: Please be more quantitative here and state approximate numbers.

2. Page 1, line 19: Add 'e.g.' before McFarquhar et al. (there are many more studies

who emphasize this aspect).

3. Page 2, 1st paragraph: The discussion on the advantages or disadvantages of remote sensing, in-situ, ground-based or airborne should be more balanced. In-situ measurements e.g. have the advantage that they are more detailed with respect to microphysical and chemical aerosol properties while they are limited in space (point measurement). Remote-sensing can cover larger areas but their retrievals often depend on assumptions and give less microphysical detail. Airborne measurements are expensive and thus not feasible for monitoring, etc. ...

4. Page 2, line 20: Are 13 references for the retrieval techniques really needed here? The same reference chain appears on page 4 again. The authors should focus on the important publications. No discussion and references on previous validation studies are given, which should be added here.

5. Page 2, line 25: Most in-situ and all monitoring measurements are usually performed at dry conditions to keep them comparable. Please add this info here.

6. Page 2, line 34: DISCOVER-AQ is not defined yet.

7. Page 3, line 3: The Zieger et al., 2010 reference is incorrect here. The lidar-in-situ comparison (using humidified nephelometer measurements) were done in Zieger et al. (2011) and in Zieger et al. (2012).

8. Page 3, line 13-14: Why was no hygroscopic adjustment necessary?

9. Page 3, Sect. 2: I would suggest to replace the title of this section ("DISCOVER-AQ") by something more descriptive (e.g. 'Campaign description' or something similar). The information on the number of profiles analysed (line 18) could also be moved to the campaign description section.

10. Page 4, Sect. 3: The section heading could also be improved here. Instead of 'LaRC HSRL-2' the authors could use 'Airborne high spectral resolution lidar' or just 'Airborne lidar measurements'. Parts of the second paragraph of Sect. 3 are repetitive from the introduction (remove it here or there). Section 3.1 is not followed by a Sect. 3.2, so I would remove this subsection heading if nothing follows or restructure.

11. Page 4, line 27: Repetition and not needed here.

12. Page 5. line 15 and Table 1: This table is not really needed and could be omitted or moved to the supplement.

13. Page 5, line 25: Does the 5 $\mu$m size cut relate to dry or to ambient RH? If the size cut relates to ambient conditions, then the effect of hygroscopic growth will influence the presented results since the actual size cut at elevated RH might much smaller. Please clarify.

14. Page 5, line 28: How was the humidified nephelometer calibrated? How did the authors determine the exact RH of the wet scattering coefficient? Were salt calibrations preformed (see e.g. recommendations given in Zieger et al., 2013)?

15. Page 5, line 16: As correctly stated, the $\gamma$-fit is one empirical fit among many. However, the real limitation is not the fit, it is rather the fact that the humidified scattering coefficients were only measured at one elevated RH due to experimental limitations (aircraft measurements). Therefore, it will remain unknown if phase transitions have occurred below 80% RH or not. In theory you can apply the $\gamma$-fit for different regimes of the humidogram (e.g. if hysteresis is present) separately (in Zieger et al., 2010, for example, the $\gamma$-fit was used to describe hysteresis effect due to deliquescent sea salt).

16. Page 5, line 22: To back-up the negligence of the absorption enhancement, the

authors should state the campaign mean and standard deviation of the single scattering albedo at this point.

17. Page 7, line 7: How were the multiple charged particles treated for the ammonium sulphate calibration of the LAS and UHSAS using the DMA?

18. Page 7, line 10-11: Please state the mean and standard deviation of the dry and wet relative humidities (preferable in the instrument section).

19. Page 7, Eq. 3: If $\bar{g}$ is not constant over the size range (as done later in the sensitivity study) then the change in number distribution has to be calculated as well. See Eq. 5 in Zieger et al. (2013).

20. Page 7, line 22: Please add that an internal mixture was assumed (correct?).

21. Sect 5.2 and Fig. 2: Every inconsistency in the in-situ measurements (e.g. particle losses) will be balanced by the retrieved refractive index. The statement, at this point, on the consistency of the in-situ measurements can only be valid if the corresponding retrieved refractive index is shown as well. This profile should be added to Fig. 2. To save space the authors could consider to show only one wavelengths for the scattering coefficient.

22. Fig. 3: The course of the hygroscopic growth factor is probably highly driven by the ambient relative humidity. For a better comparison, the RH profile should therefore be shown as well.

23. Page 10, line 24-25 and Fig. 4: The authors state that the retrievals compare well to the in-situ measurements. How were these profiles chosen? Comparing Fig. 4 with Fig. 5 it seems to be that only nice examples were cherry-picked. Therefore this sentence should be rephrased.

24. Fig. 4: The good agreement is remarkable. While the in-situ measurements of surface, volume and effective radius show clearly the same profile shape, some exceptions can be observed in the HSRL-2 retrievals. For example, in the second profile at approx. 600 m altitude (also at 1700 m), the retrieval of effective radius, surface and volume are not in correspondence. Is this due to the fact that they are independently retrieved? A consistency check could be included in the analysis (i.e. volume and surface value should give the appropriate effective radius under the assumption of spherical particles).

25. Page 10, line 27: Figure 5 is only described by one short sentence. Please be more detailed here. If the figure is not important then it should be removed.

26. Fig. 5: I find the systematic difference of the particle number concentration interesting. The HSRL-2 seems 'to see' more particles then the UHSAS. However, I would expect the UHSAS to be more sensitive to small particles which dominate the total number concentration. Is there any explanation for this? On the other hand, the surface and volume concentrations seem to agree well, while the effective radius is systematically larger for the in-situ measurements. This is surprising since the effective radius can be calculated from the surface and volume ($r_{\text{eff}} = 3V/A$; see e.g. Grainger et al., 1995) and therefore should agree well. Or is it differently defined/retrieved here? Please clarify.

27. Fig. 6: Similar to Fig. 5, I find it remarkable that the bias of the effective radius is positive with hygroscopic correction and negative without hygroscopic correction. Should it not be similar to the surface and volume concentration?

28. Page 10: The third paragraph is repetitive.

29. Page 10, line 30 and abstract: I am astonished by the fact that the bias of the total aerosol volume concentration is smaller than the on of the surface concentration. The in-situ instruments have difficulties measuring large particles (as discussed

later in the manuscript) which on the other hand determine the total aerosol volume. How can this be explained?

30. Page 11, line 15-16: Why is this interesting? Any explanations?

31. Page 11, line 19 and Fig. 8: Looking at the graphs (especially at the first panel) I would rather talk about 'good' or 'very good' agreement (and not excellent). How were these points averaged?

32. Page 12, line 17-23 and Fig. 9: Although the median bias slightly decreases if the LAS measurements are used, the IQR increases for these cases. This should be added and discussed as well.

33. Page 12, line 24-27: Particle losses was only one hypotheses among many in Zieger et al. (2011). In fact, this study also compared $3\beta + 2\alpha$ lidar measurements to in-situ recordings that were re-calculated to ambient conditions. The lidar agreed much better to the in-situ measurements than the MAX-DOAS (especially during nighttime, see Fig. 12 in Zieger et al., 2011). For the MAX-DOAS, it was hypothesized that the lowest and compared layer was overestimated due to lofted layers (e.g. caused by ammonium nitrate partitioning). In addition, the MAX-DOAS retrieval could have been influenced by horizontal gradients in aerosol concentration. Nevertheless, in this work, the influence of coarse mode particles is definitely a hot candidate for the underestimation of the in-situ data. To further investigate this, the authors could, similarly to Zieger et al. (2015), compare their in-situ optical properties to the columnar measurements of AOD (Fig. 8). Zieger et al. (2015) also found a clear underestimation of the in-situ derived AOD and hypothesized as one possible reason that coarse particles were not sufficiently sampled (e.g. being lost in the canopy or within the inlet system) due to the pronounced wavelength-dependency. The calculated fine-mode fraction could be added to Fig. 10 which would be more convincing.

[Figure]

34. No subsection is followed after 6.2.1. Therefore I would re-order and add an extra subsection or remove this heading.

35. Sect. 6.2.1 and Fig. 10: The argumentation is very speculative. The CA dataset contains much less datapoints than the TX dataset so the statistics is different. Looking at Fig. 7 again, it is hard to see a clear and significant difference between the two datasets. I would suggest to move this figure to the supplement. Alternatively, the authors could further test their hypothesis in a more convincing way, e.g. by colour-coding the points in Fig. 7 by the fine mode fraction or by plotting absolute or relative differences of the retrievals vs. the fine mode fraction of the AOD.

36. Page 13, third paragraph and Table 4: The choice and definition of the cut-off diameters is not clear to me. Sedimentation or diffusion losses should be low between 100 and 1000 nm, so I don't understand the choice of 0.7 and 0.4 $\mu$m. In addition, the hygroscopic growth factors were much larger during the campaign (up to 1.6 at elevated RH, see Fig. A1). Therefore, the particles (if the UHSAS sampled at dry conditions) where actually much larger at ambient conditions due to hygroscopic growth and the cut-off diameters should be set to values above 1 $\mu$m. The interpretation (see 4th paragraph) should then be adapted. I would interpret the sensitivity study that coarse mode particles above $\sim 1.2\,\mu$m are only really relevant for the $\beta_{1064}$-measurement. Please clarify and adapt accordingly.

37. Sect. 7 (Discussion) and Sect. 8 (Summary and conclusions): These sections are again very repetitive and often dissipate. Please focus and discuss the main findings. Both sections can be combined. The references to previous findings are missing and should be added to the discussion. The limitations of the lidar retrieval technique are not discussed or even mentioned at all in the conclusion part, which has to be added. It would be beneficial to the paper if the authors would add a short and precise outlook and recommendation part to their work.

38. Sect. 7.1: The phase transitions are important mainly for pure compounds. In the ambient atmosphere, clear and distinct phase transitions or hysteresis effects have been observed (using humidified nephelometers that look at the overall/integrated effect) mainly when sea salt was present (see e.g. Zieger et al., 2013, for an overview). Organic compounds and mixtures with other inorganic substances will most likely lead to a smooth hygroscopic behaviour without pronounced deliquescence. Figure 12 also shows a dominance of WSOC and $NO_3$ and thus makes deliquescence quite unlikely. In addition, the particles in the ambient atmosphere will most likely be on the upper branch of the hysteresis curve if they have experienced an elevated RH before the time of measurement. For the ambient optical properties, which are studied here, the authors should look and discuss the related scattering enhancement factors, which is an integrated value while HTDMA's only look at distinct (and fine mode limited) dry sizes. There is no Sect. 7.2 following 7.1, so please restructure.

39. Page 26, Table 3: The linear regression and correlation coefficients should also be given for the comparison of the microphysical parameters from Fig. 5.

40. Page 26, Table 2: The choice of biases smaller than 50% seems quite arbitrary. How is this justified?

41. Figure A1 and Sect. A1: This part is to reviewer's opinion quite important since it demonstrated the validity of the presented retrieval method for $\bar{g}$ and the good quality of the recorded in-situ data. It could be moved to the main part of the manuscript. However, it is unreasonable to show $\bar{g}$ vs. the ambient RH because the wet scattering coefficient was always measured at a constant RH (80-85 %). A comparison to $\kappa$ is therefore not appropriate since the entire curvature of $\bar{g}$ is predetermined by the here used $\gamma$-parametrization. The authors should show a distribution plot of $\bar{g}$ at RH=80-85 % (or preferable at one fixed RH by recalculating the wet scattering coefficients to one fixed RH). These values can then be

compared to literature values.

42. Sect. A2: This section is quite difficult to read and understand (again many paragraphs consisting of only one sentence). A flow chart could help here. It is not clear on why Fig. A2 has to be shown. It is probably sufficient to state that the simulations were done for similar conditions as the HSRL-2 measurements. Table A2 is hard to interpret as well. Why was the noise not added to the RH of the ambient and humidified nephelometer measurement? The same is true for the influence of the coarse mode, which might not have been sufficiently sampled. Both aspects will have a clear effect on the retrieval uncertainty (see Appendix A and Fig. A1 in Zieger et al., 2013). The sensitivity to the ambient RH is not discussed at all and should be added here. Were the ambient RH measurements of the two aircrafts compared?

43. Fig. 10: How exactly was the scaling of the volume distribution to the aerosol layer height performed? And why were two different heights (1 km and 3 km) for the two campaigns chosen. Maybe it would have been easier to just normalize all volume size distributions to 1 and then calculate the average values.

**Technical comments**

1. Page 5, line 5: Replace 'mum' by $\mu$m.

2. Throughout the manuscript: Please don't put the unit meter in italics.

3. Page 8, line 22: One web-link is probably sufficient here.

4. Fig. 2: Please use the introduced variables for the scattering and absorption coefficients (y-labels) and avoid abbreviations like Scat450 or Abs532. Instead of 'recalc' it would be better to use an abbreviation like 'retr' (retrieved).

5. Fig. 3: Please use the correct variables for the axis-labelling (see comment above). Units should be next to the numbers and not in the next row.

6. Fig. 2 and 3: To be consistent in the equations, the scattering efficiency should also depend on the dry or ambient particle diameter.

7. The section headings for 5.2 to 5.4, 6.2 are quite long and complicated. They can be shortened to be more concise (e.g. 'Retrieval of dry complex refractive index' or 'Retrieval of the effective growth factor' or 'Optical closure study').

8. Page 13, line 2: Please replace here and throughout text (where possible) $3\beta+2\alpha$ by 'extinction and backscatter coefficients'. This will improve the reading flow since the acronym is very specific for the lidar community and not known to the majority of the readers.

9. Please replace 'optical particle counters' by 'optical particle size spectrometers'. UHSAS and LAS are not just counting particles like a CPC, they also size them.

10. Fig. 4: Please add that these profiles are for the fine mode fraction only.

11. Fig. 8: Units missing in the statistics text blocks.

12. Fig. 11: Define SZD at the beginning of the caption.

13. Page 17, line 9: $\kappa$ can range according to Petters and Kreidenweis (2007) up to 1.3 (sea spray).

14. Page 17, line 29: The correct formula for ammonium sulphate is $(NH_4)_2SO_4$

15. Sect. A1: Please harmonize the variable names (i.e. the refractive index is given in different ways).

16. Fig. 1: Replace 'King Air' by 'B-200' as shown in the figure (or vice versa).

17. AOT and AOD are not used in the same way throughout the manuscript. For example, in Fig. 8 it is AERONET AOT and in Fig. 10 it is AOD for the fine fraction. Please harmonize.

**References**

Grainger R., Lambert A., Rodgers C., Taylor F., and Deshler T., Stratospheric aerosol effective radius, surface area and volume estimated from infrared measurements, *J. Geophys. Res.*, 100(D8), 16507–16518, 1995.

Petters M. and Kreidenweis S., A single parameter representation of hygroscopic growth and cloud condensation nucleus activity, *Atmos. Chem. Phys.*, 7(8), 1961–1971, doi: 10.5194/acp-7-1961-2007, 2007.

Zieger P., Aalto P.P., Aaltonen V., Äijälä M., Backman J., Hong J., Komppula M., Krejci R., Laborde M., Lampilahti J., de Leeuw G., Pfüller A., Rosati B., Tesche M., Tunved P., Väänänen R., and Petäjä T., Low hygroscopic scattering enhancement of boreal aerosol and the implications for a columnar optical closure study, *Atmos. Chem. Phys.*, 15(13), 7247–7267, doi:10.5194/acp-15-7247-2015, 2015.

Zieger P., Fierz-Schmidhauser R., Gysel M., Ström J., Henne S., Yttri K., Baltensperger U., and Weingartner E., Effects of relative humidity on aerosol light scattering in the Arctic, *Atmos. Chem. Phys.*, 10(8), 3875–3890, doi:10.5194/acp-10-3875-2010, 2010.

Zieger P., Fierz-Schmidhauser R., Weingartner E., and Baltensperger U., Effects of relative humidity on aerosol light scattering: results from different European sites, *Atmos. Chem. Phys.*, 13(21), 10609–10631, doi:10.5194/acp-13-10609-2013, 2013.

Zieger P., Kienast-Sjögren E., Starace M., v. Bismarck J., Bukowiecki N., Baltensperger U., Wienhold F., Peter T., Ruhtz T., Collaud Coen M., Vuilleumier L., Maier O., Emili E., Popp C., and Weingartner E., Spatial variation of aerosol optical properties around the high-alpine site Jungfraujoch (3580 m a.s.l.), *Atmos. Chem. Phys.*, 12, 7231–7249, doi: 10.5194/acp-12-7231-2012, 2012.

Zieger P., Weingartner E., Henzing J., Moerman M., de Leeuw G., Mikkilä J., Ehn M., Petäjä T., Clémer K., van Roozendael M., Yilmaz S., Frieß U., Irie H., Wagner T., Shaiganfar R., Beirle S., Apituley A., Wilson K., and Baltensperger U., Comparison of ambient aerosol extinction

coefficients obtained from in-situ, MAX-DOAS and LIDAR measurements at Cabauw, *Atmos. Chem. Phys.*, 11(6), 2603–2624, doi:10.5194/acp-11-2603-2011, 2011.

---

## Referee Comment (RC2) · Anonymous Referee #2 · 22 Jun 2016

The title and the abstract describe very well what the intent of the manuscript is. The abstract is very well written. The work overall is very thorough and should definitely be published.

This reviewer believes the authors' intent is to ultimately validate the HSRL-2 retrievals so they can be applied globally to a similar spaceborne lidar system which is being considered for the ACE (Aerosol-Cloud-Ecosystem). The strong point being that these measurements will be representing ambient conditions (i.e. the conditions relevant for radiative transfer) and can be global, whereas the in situ measurements (while more

detailed) are neither global nor taken at ambient conditions. This reviewer feels it would be important to make this point in the introduction of the manuscript.

Many corrections that are well described in the manuscript need to be applied to the in-situ data before they can be compared to the in-situ data. The quantitative comparisons and trends are very impressive and revealing. It would be helpful to mention how well extinction can actually be measured with in-situ data. Schmid et al. (2006) some years ago have arrived at such an estimate by looking at a large number of campaigns.

This reviewer agrees with what seems to be one of the main conclusions of the manuscript. "Regardless of the number of uncertainties that can affect studies like this one, we have demonstrated that the HSRL-2 retrievals of fine mode aerosol size parameters are well correlated to in situ measurements. Further work is still necessary in order to more effectively quantify the net effects of such uncertainties in comparison studies of this kind." In other words, are more comparisons needed to arrive at a definitive validation of the HSRL-2 aerosol microphysical retrievals? AOD comparisons with ground based sunphotometry have also been shown and look impressive. Should this be emphasized more?

Specific Comments:

Page 1, Line 19: This reviewer believes the McFarquhar reference is missing. These references are just examples so should be prefaced with "e.g."

Page 2, Line 20: This reviewer suggests using "focused on" instead of "in"

Page 3, Line 8: This reviewer suggests using "non-representative" instead of "unrepresentative"

Page 3, Lines 10-14: Text makes it sound like only 2 spirals were flown in TCAP. Which sounds incorrect. Please also explain that HSRL-2 was only deployed in phase 1 of TCAP. It would be nice to mention the in situ aircraft by name (DOE G-1) and perhaps use the corresponding reference (Schmid et al., 2014). Definitely must explain

why hygroscopic adjustments were not necessary. Also please quantify the level of agreement instead of just saying "good agreement". It might also be worth mentioning comparisons done in TCAP Phase I between HSRL-2 and another remote sensing method (4STAR, Shinozuka et al., 2013a, b).

Page 4, Line 4: Missing "the" before NASA

Page 5: Line 5: micron symbol got lost

Page 6, Line 25: Please specify that this is the airborne version of UHSAS mounted in the free airstream. Who good is sizing of UHSAS near 1 micron?

Page 8, Line 22: "most" leads me to ask the question "why not all" data are publicly available.

Page 9, Line 8: The size range of UHSAS is a subset of the size range the Neph and PSAP sees unless a size cut-off of 1 micron was used which is not mentioned. You need the LAS data for the larger sizes. Have you done a cumulative scattering calculation as in Kassianov et al. to see what percentage of scattering the UHSAS is missing?

Page 11, Line 25: At this point in the manuscript this reviewer was very surprised to learn that only the fine mode had been considered. Finding this out on page 11 is too late and a bit frustrating.

Page 12, Line 4: At this point you need to say that the cut-off is likely not sharp at exactly 5 microns. Rather it probably has an S-shape. Some particles larger than the cut-off will still make it through the inlet, whereas some that are smaller than the cut-off won't. So is 5 micron the 50% efficiency point?

Page 12, Line 4: Very late in the manuscript (p.12) to say that supermicron data from LAS were not available for DAQ TX.

Page 12, Line 11. Good to see this improvement!

Page 12, Line 14. Unclear. Increase of what?

Page 20, Line 25. Some strange symbols

Figure 2: Should use a different symbol than alpha as this is used already for extinction. In 3 beta +2 alpha.

References

Schmid B., R. Ferrare, C. Flynn, R. Elleman, D. Covert, A. Strawa, E. Welton, D. Turner, H. Jonsson, J. Redemann, J. Eilers, K. Ricci, A. G. Hallar, M. Clayton, J. Michalsky, A. Smirnov, B. Holben, J. Barnard (2006). How well do state-of-the-art techniques measuring the vertical profile of tropospheric aerosol extinction compare? J. Geophys. Res. 111, D05S07, doi:10.1029/2005JD005837.

Schmid B., J. M. Tomlinson, J. M. Hubbe, J. M. Comstock, F. Mei, D. Chand, M. S. Pekour, C. D. Kluzek, E. Andrews, S.C. Biraud, G. M. McFarquhar (2014). The DOE ARM Aerial Facility. Bull. Amer. Meteor. Soc., 95(5), 723–742, doi: 10.1175/BAMS-D-13-00040.1

Shinozuka Y., R.R. Johnson, C.J. Flynn, P.B. Russell, B. Schmid, J. Redemann, S.E. Dunagan, C.D. Kluzek, J.M. Hubbe, M. Segal-Rosenheimer, J.M. Livingston, T.F. Eck, R. Wagener, L. Gregory, D. Chand, L.K. Berg, R.R. Rogers, R.A. Ferrare, J.W. Hair, and C.A. Hostetler (2013). Hyperspectral aerosol optical depths from TCAP flights. J. Geophys. Res. Atmos., 118, doi:10.1002/2013JD020596.

Shinozuka Y., R.R. Johnson, C.J. Flynn, P.B. Russell, B. Schmid, J. Redemann, S.E. Dunagan, C.D. Kluzek, J.M. Hubbe, M. Segal-Rosenheimer, J.M. Livingston, T.F. Eck, R. Wagener, L. Gregory, D. Chand, L.K. Berg, R.R. Rogers, R.A. Ferrare, J.W. Hair, and C.A. Hostetler (2014). Correction to "Hyperspectral aerosol optical depths from TCAP flights", J. Geophys. Res. Atmos., 119, 1692–1693, doi:10.1002/jgrd.51089.

---

## Referee Comment (RC3) · Anonymous Referee #3 · 26 Jun 2016

The study describes the retrieval products and measurements of HSRL-2, an airborne multi-wavelength lidar, from two phases of the DISCOVER-AQ experiment and evaluates them with accompanying in situ aerosol measurements. The data products discussed are particle number concentration, surface area, volume, effective radius, extinction and backscattering.

As far as I know, lidar-based studies hitherto either stayed largely in the domain of optical properties or explored microphysical retrievals with a small number, if any, of measurements. The present study distinguishes itself from them by providing as many

as >700 data points of microphysical retrieval products. The fairly thorough analysis will prove useful if, as it seems likely, the HSRL-2 microphysical retrieval products are to be used for the studies of aerosol effects on climate and air quality.

I have one issue with the data analysis.

I suspect the real part of dry refractive index is systematically overestimated. That is because of the discrepancy in particle size between the two sets of measurement being compared: The submicron particles that the UHSAS observed are held accountable for the extinction by the particles up to 5 um that the nephelometer and PSAP observed.

To reduce the systematic error, one could compare size distribution and extinction for an identical size range. An impactor is commonly used to pass particles under 1 um aerodynamic diameter. Its passing efficiency modeled for geometric diameters (see, for example, paragraph [21] of Howell et al., 2006) allows adjustment of the measured size distribution for the particles behind it. Optimize the dry refractive index for the adjusted dry size distribution and the scattering and absorption measured behind the impactor.

The overestimate in refractive index, which I think should be noted in the manuscript, has implications. It invites a systematic bias in the calculated extinction and backscattering except the extinction in the vicinity of the nephelometer and PSAP wavelengths (i.e., 532 nm extinction). So the behind-the-impactor retrieval may help explain the systematic biases shown in Figure 7. This possibility makes it worth trying even if the random error is to be magnified for the smaller coefficients and the uncertainty in the impactor passing efficiency.

Minor suggestions.

Page 1. Line 1. Insert "and" after "radii".

Page 2. Line 18. Replace the slash after dsm with a period.

Page 3. Line 14. "not necessary". Why? Low ambient RH?

[Figure]

Page 3. Line 16. Remove "of more than 700 lidar retrievals" because it is said in line 18.

Page 4. Line 21. Insert ", the latter" after "California".

Page 5. Line 5. Use the Greek letter instead of mu.

Page 5. Line 11. The first sentence is unclear. Is it necessary?

Page 6. Line 18. Are these wavelengths correct?

Page 6. Line 29. Replace "proportional" with "related".

Page 8. Line 13. The vertical resolution of 5 m corresponds to ~1s for typical aircraft vertical speeds. But, while the TSI nephelometer records every second, it does not resolve scattering coefficient for each second. The residence time of particles in the TSI nephelometer is closer to 5s under typical flow rates.

Page 10. Line 16. Is "approximately" necessary? Also Page 16. Line 8.

Page 10. Line 29. Replace "measuremets" with "measurements".

Page 11. Line 2. Replace "sensitive" with "sensitivity".

Page 13. Line 15. Replace "seem" with "seems".

Page 13. Line 28-30. Isn't this because the particles sampled in California were somewhat smaller than those in Texas, as implied in Figure A2? Smaller particles are less prone to inlet loss. Can you show the bias for the 532 nm extinction as a function of the Angstrom exponent? The 532 nm extinction is a good choice here because it should be barely affected by the refractive index bias mentioned above.

Page 16. Line 2. What does "a preliminary assessment . . ." refer to?

Page 18. Line 32. Make "I" small.

Figure 2. Note the particle size range for the dN/dlogD (< 1um) and the measured

scattering and absorption (< 5um).

Figure 3. Make the "O" as large as "H" in the upper right box.

Figure 4. Indicate that the values refer to the fine-mode only. Perhaps also for Figure 5 and 6.

Figure 6. Should "q1+1.5xIQR" read "q3+1.5xIQR"? Also, what is the significance of 1.5xIQR? Why is this expression used instead of another set of percentiles like 5% and 95%?

Reference

S. G. Howell, A. D. Clarke, Y. Shinozuka, V. Kapustin, C. S. McNaughton, B. J. Huebert, S. J. Doherty, and T. L. Anderson, 'Influence of Relative Humidity Upon Pollution and Dust During Ace-Asia: Size Distributions and Implications for Optical Properties', Journal of Geophysical Research-Atmospheres, 111 (2006), D06205, doi:10.1029/2004JD005759.

---

## Referee Comment (RC4) · Anonymous Referee #4 · 6 Jul 2016

General comments: Sawamura and coauthors present a comparison of aerosol microphysical and optical properties obtained from airborne in-situ and lidar measurements. The dataset is unique, with over 700 vertically resolved profiles of aerosol microphysical properties, which makes this comparison of techniques more robust than those previously published in the literature. The topic is sound and of interest for the scientific community. However, there are many clarifications and modifications that have to be adequately addressed before publication in ACP. The overall manuscript structure is not well defined, which makes the manuscript difficult to read and the focus is often lost. All the retrievals explanations should be included in the methods section (avoid

methodological aspects in the results section). In addition, small and individual sentences make up a paragraph. These topic-related sentences could be combined in a single paragraph to make the reading more fluent. There are too many tables and figures, I would suggest to keep just the important ones.

Specific comments

Page 2, line 20-34: Since the main objective of the paper is the evaluation of lidar retrievals with airborne insitu data, some discussion should be included here about what have been done in the past. For example, Granados-Muñoz et al. (2016) compared aerosol volume concentrations retrieved from Lidar measurements with airborne in-situ size-distributions. The paragraph in page 3, line 10 about Müller et al. (2014) could be moved here for consistency. Granados-Muñoz et al., A comparative study of aerosol microphysical properties retrieved from ground-based remote sensing and aircraft in situ measurements during a Saharan dust event, Atmos. Meas. Tech., 9, 1113–1133, doi:10.5194/amt-9-1113-2016, 2016.

Page 3, line 14: Why hygroscopic adjustments were not necessary?

Page 3: sections 2, 3 and 4 headings could be improved. . .

Page 5: aerosol properties are measured in situ onboard the P-3B aircraft. . .

Page 5, line 5: replace "mum" by "$\mu$m".

Page 6, line 1: How is the RH<40% achieved? nafion dryer?

Page 6, line 1: State the actual RH inside the dry nephelometer (mean and std)

Page 6: How well do the two nephelometers compare when measuring at RH<40%?

Page 6, line 2: I see one major problem here, and it is the calculation of the gamma parameter with only two RHs. Additional discussion about the errors and uncertainties introduced with this approach should be included here. This will be very important for deliquescent aerosols.

Page 6, line 13: The authors refer to aerosol backscattering instead of scattering, right?

Page 6, Line 18: Are the PSAP wavelengths correct?

Page 6, line 21: A reference supporting the non-hygroscopic enhancement in the absorption coefficient should be included here.

Page 6, line 24-Page 7, line 8: The UHSAS and LAS instruments overlap in the range 0.09-1 $\mu$m, have the authors compared the size distributions in the overlapping region? Have these instruments been intercompared with more robust instrument like SMPS?

Page 7, line 10: only at 550 nm?

Page 7, lines 14-15: This is a repetition. . .

Page 7, lines 23-28: These two paragraphs are confusing; the first one ends with ". . . mdry values are unknown." and the second one starts with "Once mdry is determined. . .". How is mdry determined? It is not said until section 5.2 in page 8! This section is jumpy and messy. . .

Page 8, line 30: nephelometer

Section 5.2: why the size distribution measured with LAS is not used? The nephelometers are sampling up to 5 $\mu$m (inlet cut-off) but the size distribution used is restricted to diameters < 1 $\mu$m.

Section 5.2: The UHSAS is calibrated using AS (m = 1.53) as stated previously, and then this AS-calibrated size distribution is used to retrieve mdry. Thus, from the reviewer's point of view, the mdry obtained should be seen as an "effective" refractive index able to reproduce your optical measurements. Therefore, the consistency of the insitu measurements (as stated in page 9, line 10) is not demonstrated.

Figure 3: it would be nice to see the ambient RH profile as well.

Section 5.2 and Figure 3: The authors should keep in mind that the scattering coeffi-

cient at ambient conditions is not a measured variable, is a calculated one (and likely with a high uncertainty due to the gamma fit).

Page 9, last paragraph: the symbols for the extinction and backscatter coefficient should be introduced earlier. The computed backscatter coefficient, refer to 180° or to integrated hemispheric backscatter?

Page 10, 3rd paragraph: repetition

Page 10, line 17: why the coarse mode is not studied? The notation in graphs and in the text should make clear that the surface and volume concentrations refer only to the fine mode.

Figure 4: Why these profiles have been shown?

Figure 5: In-situ also refers to fine mode only

Page 10, line 25: do the authors have any explanation about the different agreement found for the surface and volume concentrations?

Page 10, line 27: Any explanations about why the bias was larger in California than in Texas? I would suggest presenting the results combined with the discussion.

Section 6.2: It would make more sense to compare the measured in-situ extinction coefficient (scattering + absorption) with the HSRL-2 extinction coefficient, rather than the retrieved insitu extinction coefficient. . .

Figure 8 can be omitted.

Section 7: As mentioned before, it would be better to present the results together with their discussion.

Page 13, line 23: initially?

---

## Author Comment (AC1) · 1 Sep 2016

**Author comments to Referee #1 (RF1)**

*General comments:*

The authors would like to thank RF1 for his/her extensive and very thorough review of our manuscript. Your suggestions were carefully considered and addressed.

*Response to detailed comments:*

1. Page 1, line 10-13: Please be more quantitative here and state approximate numbers.

    Fixed.

2. Page 1, line 19: Add 'e.g.' before McFarquhar et al. (there are many more studies C2 who emphasize this aspect).

    Fixed.

3. Page 2, 1st paragraph: The discussion on the advantages or disadvantages of remote sensing, in-situ, ground-based or airborne should be more balanced. In-situ measurements e.g. have the advantage that they are more detailed with respect to microphysical and chemical aerosol properties while they are limited in space (point measurement). Remote-sensing can cover larger areas but their retrievals often depend on assumptions and give less microphysical detail. Airborne measurements are expensive and thus not feasible for monitoring, etc. ...

    The Introduction section was modified accordingly.

4. Page 2, line 20: Are 13 references for the retrieval techniques really needed here? The same reference chain appears on page 4 again. The authors should focus on the important publications. No discussion and references on previous validation studies are given, which should be added here.

    References were revised and repetition of references was removed. We added references of other validation studies as suggested.

5. Page 2, line 25: Most in-situ and all monitoring measurements are usually performed at dry conditions to keep them comparable. Please add this info here.

    This is mentioned a couple of paragraphs later, therefore the authors felt it would be repetitive to add this information here.

6. Page 2, line 34: DISCOVER-AQ is not defined yet.

    We moved the pertinent paragraph to the next section.

7. Page 3, line 3: The Zieger et al., 2010 reference is incorrect here. The lidarin-situ comparison (using humidified nephelometer measurements) were done in Zieger et al. (2011) and in Zieger et al. (2012).

    The references were corrected.

8. Page 3, line 13-14: Why was no hygroscopic adjustment necessary?

According to the authors of Mueller et al, 2014, it was thought that the relative humidity during the cases presented in their study was low enough and therefore hygroscopic adjustments could be neglected. In our current study, the need for hygroscopicity adjustments is emphasized.

9. Page 3, Sect. 2: I would suggest to replace the title of this section (”DISCOVERAQ”) by something more descriptive (e.g. ’Campaign description’ or something similar). The information on the number of profiles analysed (line 18) could also be moved to the campaign description section.

Section title was changed and recommended changes were applied.

10. Page 4, Sect. 3: The section heading could also be improved here. Instead of ’LaRC HSRL-2’ the authors could use ’Airborne high spectral resolution lidar’ or just ’Airborne lidar measurements’. Parts of the second paragraph of Sect. 3 are repetitive from the introduction (remove it here or there). Section 3.1 is not followed by a Sect. 3.2, so I would remove this subsection heading if nothing follows or restructure.

Section was renamed and subsections removed.

11. Page 4, line 27: Repetition and not needed here.

Repetition removed.

12. Page 5. line 15 and Table 1: This table is not really needed and could be omitted or moved to the supplement.

Table was moved to supplement.

13. Page 5, line 25: Does the 5 μm size cut relate to dry or to ambient RH? If the size cut relates to ambient conditions, then the effect of hygroscopic growth will influence the presented results since the actual size cut at elevated RH might much smaller. Please clarify.

The sentenced was changed to reflect that the cut-off size was related to dry conditions.

14. Page 5, line 28: How was the humidified nephelometer calibrated? How did the authors determine the exact RH of the wet scattering coefficient? Were salt calibrations preformed (see e.g. recommendations given in Zieger et al., 2013)?

The humidified nephelometer is calibrated in the same manner as the dry nephelometer by measuring the scattering coefficient of dry $CO_2$ gas, whose scattering properties are known in the literature. This procedure is standard and is described in the TSI manual.  The exact RH was determined from three different relative humidity probes located at the inlet, exhaust, and within the nephelometer flow cell. Ammonium sulfate was used to test the f(RH) for the nephelometers.

15. Page 5, line 16: As correctly stated, the γ-fit is one empirical fit among many. However, the real limitation is not the fit, it is rather the fact that the humidified scattering coefficients were only measured at one elevated RH due to experimental limitations (aircraft measurements). Therefore, it will remain unknown if phase transitions have occurred below 80% RH or not. In theory you can apply the γ-fit for different regimes of the humidogram (e.g. if hysteresis is present) separately (in Zieger et al., 2010, for example, the γ-fit was used to describe hysteresis effect due to deliquescent sea salt).

*The original paragraph has been modified, and the experimental limitations with respect to RHwet measurements are discussed later in the manuscript, along with the discussion of chemical composition of the aerosols observed in Texas (see last paragraph of Section 6.4 of the revised manuscript).*

16. Page 5, line 22: To back-up the negligence of the absorption enhancement, the authors should state the campaign mean and standard deviation of the single scattering albedo at this point.

*Mean values, standard deviations, and 90th percentiles of dry single scattering albedo measurements at 550 nm during DAQ CA and TX were stated.*

17. Page 7, line 7: How were the multiple charged particles treated for the ammonium sulphate calibration of the LAS and UHSAS using the DMA?

*Multiply charged particles transmitted through the DMA appear as distinct, smaller peaks in the optical size distributions of the LAS and UHSAS. These distinct peaks were ignored — only the singly-charged DMA peak was used.*

18. Page 7, line 10-11: Please state the mean and standard deviation of the dry and wet relative humidities (preferable in the instrument section).

*Mean and standard deviations of dry and wet RH were added to the in situ instruments section.*

19. Page 7, Eq. 3: If $\bar{g}$ is not constant over the size range (as done later in the sensitivity study) then the change in number distribution has to be calculated as well. See Eq. 5 in Zieger et al. (2013).

*All size distributions in the sensitivity study were normalized to 1 at dry conditions. Assuming the hygroscopic growth does not change the number of particles, the wet size distributions were renormalized to 1 once the growth factors were applied.*

20. Page 7, line 22: Please add that an internal mixture was assumed (correct?).

*Fixed.*

21. Sect 5.2 and Fig. 2: Every inconsistency in the in-situ measurements (e.g. particle losses) will be balanced by the retrieved refractive index. The statement, at this point, on the consistency of the in-situ measurements can only be valid if the corresponding retrieved refractive index is shown as well. This profile should be added to Fig. 2. To save space the authors could consider to show only one wavelengths for the scattering coefficient.

*Figure 2 was modified accordingly.*

22. Fig. 3: The course of the hygroscopic growth factor is probably highly driven by the ambient relative humidity. For a better comparison, the RH profile should therefore be shown as well.

*Figure 3 was modified accordingly.*

23. Page 10, line 24-25 and Fig. 4: The authors state that the retrievals compare well to the in-situ measurements. How were these profiles chosen? Comparing Fig. 4 with Fig. 5 it seems to be that only nice examples were cherry-picked. Therefore this sentence should be rephrased.

The profiles were chosen qualitatively based on how complete the profiles were – both in terms of the in situ and the lidar retrievals. Due to the small number of points in any particular profile obtained in California (due to the shallow PBL), none of those cases were used as examples. The correspondence between Figure 4 and Figure 5 can be seen if looked carefully. Surface area and volume concentrations show better agreement while effective radii are slightly underestimated by the lidar retrievals, in comparison to the in situ measurements (adjusted to hygroscopicity). The sentence in question was not modified because the authors believe that it does not misrepresent the comparison.

24. Fig. 4: The good agreement is remarkable. While the in-situ measurements of surface, volume and effective radius show clearly the same profile shape, some exceptions can be observed in the HSRL-2 retrievals. For example, in the second profile at approx. 600 m altitude (also at 1700 m), the retrieval of effective radius, surface and volume are not in correspondence. Is this due to the fact that they are independently retrieved? A consistency check could be included in the analysis (i.e. volume and surface value should give the appropriate effective radius under the assumption of spherical particles).

The number, surface area, and volume concentrations are not independent retrievals. The effective radius is calculated from the surface area and volume concentrations, therefore, the correspondence is implied. In the examples provided by the reviewer it is not correct to state that the retrievals of effective radius, and surface area and volume concentrations are not in correspondence. At 600 m, for instance, the surface area retrieval is about 330 um^2/cm^3, the volume is about 11 um^3/cm^3, and therefore reff = 3V/S = 0.10 um (this data point in particular cannot be clearly visualized in Figure 4 but part of the error bar can be seen). At 1.7km the surface area retrieval is about 47um^2/cm^3 and the volume is about 3.9 um^3/cm^3 which results in reff = 0.25 um.

25. Page 10, line 27: Figure 5 is only described by one short sentence. Please be more detailed here. If the figure is not important then it should be removed.

Fixed.

26. Fig. 5: I find the systematic difference of the particle number concentration interesting. The HSRL-2 seems 'to see' more particles then the UHSAS. However, I would expect the UHSAS to be more sensitive to small particles which dominate the total number concentration. Is there any explanation for this? On the other hand, the surface and volume concentrations seem to agree well, while the effective radius is systematically larger for the in-situ measurements. This is surprising since the effective radius can be calculated from the surface and volume (reff = 3V /A; see e.g. Grainger et al., 1995) and therefore should agree well. Or is it differently defined/retrieved here? Please clarify.

The reviewer is correct that the lidar is less sensitive to small particles; the relative lack of sensitivity leads to inaccuracies in the retrieved number concentrations of very small particles. Since this is a retrieval, the inaccuracy could lead to either increased or decreased number in the less-sensitive size range. The definition of effective radius used in this study is the same as the one mentioned by the reviewer. The larger apparent disagreement in effective radius is not inconsistent. The surface and volume errors are correlated and the effective radius is a ratio. Also, the surface and volume profiles have more vertical variability. An error in S or V may look relatively small compared to the relatively more constant effective radius profile. You can see that they are consistent by looking

carefully at errors in particular layers. The effective radius is systematically larger or smaller for large segments but not the entire profiles. In those regions, either S is systematically smaller or V is systematically larger (but these are less obvious).

27. Fig. 6: Similar to Fig. 5, I find it remarkable that the bias of the effective radius is positive with hygroscopic correction and negative without hygroscopic correction. Should it not be similar to the surface and volume concentration?

In fact, is the other way around, as can be seen in Figure 6. What we observed is that the bias in the effective radius was negative with hygroscopic correction, and positive without the hygroscopic correction. The bias for the effective radius is not necessarily similar to the bias in surface and volume concentration because of the effective growth factor adjustment.

28. Page 10: The third paragraph is repetitive.

Fixed.

29. Page 10, line 30 and abstract: I am astonished by the fact that the bias of the total aerosol volume concentration is smaller than the on of the surface concentration. The in-situ instruments have difficulties measuring large particles (as discussed later in the manuscript) which on the other hand determine the total aerosol volume. How can this be explained?

The coarse mode volume that is not measured by the in situ measurements are not included in the comparison because we only considered the fine mode retrievals. But the reviewer's intuition agrees with what has been observed in the evaluation of the retrieval algorithm using synthetic data (Table 2). It is possible that this discrepancy observed in our study is due to the uncertainties in the retrieval of g which have not been independently assessed. Figure 6 shows that without the hygroscopic correction, the biases are smaller for surface area than for volume concentrations.

30. Page 11, line 15-16: Why is this interesting? Any explanations?

It is interesting because later on, on page 13 (lines 13-16), when summarizing the results of the simulation described in Section 6.2.1, we also observe that one of the cutoff effects seems to be that the extinction is better reproduced than the backscatter coefficient. The manuscript will be restructured in order to make this connection clearer.

31. Page 11, line 19 and Fig. 8: Looking at the graphs (especially at the first panel) I would rather talk about 'good' or 'very good' agreement (and not excellent). How were these points averaged?

"Excellent" was replaced by "very good". As described in that same paragraph, we used the HSRL-2 measurements obtained within 2.5 km and 10 minutes from the AERONET measurements for the average.

32. Page 12, line 17-23 and Fig. 9: Although the median bias slightly decreases if the LAS measurements are used, the IQR increases for these cases. This should be added and discussed as well.

Figure 9 was outdated. The authors apologize for the confusion. The figure and the discussion have been updated.

33. Page 12, line 24-27: Particle losses was only one hypotheses among many in Zieger et al. (2011). In fact, this study also compared 3β+2α lidar measurements to in-situ recordings that were re-calculated to ambient conditions. The lidar agreed much better to the in-situ measurements than the MAX-DOAS (especially during nighttime, see Fig. 12 in Zieger et al., 2011). For the MAX-DOAS, it was hypothesized that the lowest and compared layer was overestimated due to lofted layers (e.g. caused by ammonium nitrate partitioning). In addition, the MAXDOAS retrieval could have been influenced by horizontal gradients in aerosol concentration. Nevertheless, in this work, the influence of coarse mode particles is definitely a hot candidate for the underestimation of the in-situ data. To further investigate this, the authors could, similarly to Zieger et al. (2015), compare their in-situ optical properties to the columnar measurements of AOD (Fig. 8). Zieger et al. (2015) also found a clear underestimation of the in-situ derived AOD and hypothesized as one possible reason that coarse particles were not sufficiently sampled (e.g. being lost in the canopy or within the inlet system) due to the pronounced wavelength-dependency. The calculated fine-mode fraction could be added to Fig. 10 which would be more convincing.

> This paragraph was modified. We worked on the reviewer's suggestion of comparing integrated in situ measurements of extinction to AERONET AOT measurements and new figures were added to the manuscript accordingly.

34. No subsection is followed after 6.2.1. Therefore I would re-order and add an extra subsection or remove this heading.

> Sub-subsection 6.2.1 was changed to subsection 6.3.

35. Sect. 6.2.1 and Fig. 10: The argumentation is very speculative. The CA dataset contains much less datapoints than the TX dataset so the statistics is different. Looking at Fig. 7 again, it is hard to see a clear and significant difference between the two datasets. I would suggest to move this figure to the supplement. Alternatively, the authors could further test their hypothesis in a more convincing way, e.g. by colour-coding the points in Fig. 7 by the fine mode fraction or by plotting absolute or relative differences of the retrievals vs. the fine mode fraction of the AOD.

> The statistics presented in Figure 10 were calculated using AERONET data only, and therefore do not correspond to the same statistics presented in Figure 7. The AOD fine fraction box plot presented in Figure 10 corresponds to the order of 19,000 data points for California and 21,000 for Texas.

36. Page 13, third paragraph and Table 4: The choice and definition of the cut-off diameters is not clear to me. Sedimentation or diffusion losses should be low between 100 and 1000 nm, so I don't understand the choice of 0.7 and 0.4 μm. In addition, the hygroscopic growth factors were much larger during the campaign (up to 1.6 at elevated RH, see Fig. A1). Therefore, the particles (if the UHSAS sampled at dry conditions) where actually much larger at ambient conditions due to hygroscopic growth and the cut-off diameters should be set to values above 1 μm. The interpretation (see 4th paragraph) should then be adapted. I would interpret the sensitivity study that coarse mode particles above ~ 1.2 μm are only really relevant for the β1064-measurement. Please clarify and adapt accordingly.

> The simulations presented in Section 6.2.1 had the objective to show how much of each component of a 3+2 dataset would be reproduced at varying cut-off diameters. In the simulations we probe cutoff diameters from 11 nm to 21 um. The choice of reporting the reproduced fractions at cutoff

diameters of 0.4, 0.7 and 1.2 um were chosen arbitrarily. As correctly stated, the hygroscopic factor values ranged from 1 to about 1.6. The value 1.2 for g was based on the median g values for both campaigns, which were both about 1.2.

37. Sect. 7 (Discussion) and Sect. 8 (Summary and conclusions): These sections are again very repetitive and often dissipate. Please focus and discuss the main findings. Both sections can be combined. The references to previous findings are missing and should be added to the discussion. The limitations of the lidar retrieval technique are not discussed or even mentioned at all in the conclusion part, which has to be added. It would be beneficial to the paper if the authors would add a short and precise outlook and recommendation part to their work.

The sections were modified.

38. Sect. 7.1: The phase transitions are important mainly for pure compounds. In the ambient atmosphere, clear and distinct phase transitions or hysteresis effects have been observed (using humidified nephelometers that look at the overall/integrated effect) mainly when sea salt was present (see e.g. Zieger et al., 2013, for an overview). Organic compounds and mixtures with other inorganic substances will most likely lead to a smooth hygroscopic behaviour without pronounced deliquescence. Figure 12 also shows a dominance of WSOC and NO3 and thus makes deliquescence quite unlikely. In addition, the particles in the ambient atmosphere will most likely be on the upper branch of the hysteresis curve if they have experienced an elevated RH before the time of measurement. For the ambient optical properties, which are studied here, the authors should look and discuss the related scattering enhancement factors, which is an integrated value while HTDMA's only look at distinct (and fine mode limited) dry sizes. There is no Sect. 7.2 following 7.1, so please restructure.

This discussion has been updated.

39. Page 26, Table 3: The linear regression and correlation coefficients should also be given for the comparison of the microphysical parameters from Fig. 5.

A table was added with the correlation coefficients and fit parameters for the microphysical properties in the Supplemental Material.

40. Page 26, Table 2: The choice of biases smaller than 50% seems quite arbitrary. How is this justified?

50% is a number that has been used as a worst case benchmark when testing the lidar retrieval algorithms. This threshold will most likely change in the near future to take into account the measurement requirements set by ACE (Aerosol-Cloud-Ecosystems).

41. Figure A1 and Sect. A1: This part is to reviewer's opinion quite important since it demonstrated the validity of the presented retrieval method for g̅ and the good quality of the recorded in-situ data. It could be moved to the main part of the manuscript. However, it is unreasonable to show g̅ vs. the ambient RH because the wet scattering coefficient was always measured at a constant RH (80-85 %). A comparison to κ is therefore not appropriate since the entire curvature of g̅ is predetermined by the here used γ-parametrization. The authors should show a distribution plot of g̅ at RH=80-85 % (or preferable at one fixed RH by recalculating the wet scattering coefficients to one fixed RH). These values can then be compared to literature values.

> The discussion on growth factors have been modified, and the suggested distribution plot was added to the manuscript in the methodology section.

42. Sect. A2: This section is quite difficult to read and understand (again many paragraphs consisting of only one sentence). A flow chart could help here. It is not clear on why Fig. A2 has to be shown. It is probably sufficient to state that the simulations were done for similar conditions as the HSRL-2 measurements. Table A2 is hard to interpret as well. Why was the noise not added to the RH of the ambient and humidified nephelometer measurement? The same is true for the influence of the coarse mode, which might not have been sufficiently sampled. Both aspects will have a clear effect on the retrieval uncertainty (see Appendix A and Fig. A1 in Zieger et al., 2013). The sensitivity to the ambient RH is not discussed at all and should be added here. Were the ambient RH measurements of the two aircrafts compared?

> Figure A2 was removed. In the "noise-added scenario" the 20% error in the ambient scattering coefficient is assumed to account for the errors of all variables used to calculate the scattering, i.e. gamma, dry scattering coefficient, and ambient RH. This simulation is a best case scenario assessment of how the in situ retrieval algorithm would perform if all in situ instruments were consistent among themselves in terms of cutoff diameter. A note has been added in the summary section of the revised version. It is unclear whether there were RH measurements onboard the B200. But even if there were, such comparison would not be relevant in our study since the B-200 was flown at a constant altitude (approx. 8.5 km, i.e. higher than the P3B) and the measurements at the B200 aircraft level are not really used in this study.

43. Fig. 10: How exactly was the scaling of the volume distribution to the aerosol layer height performed? And why were two different heights (1 km and 3 km) for the two campaigns chosen. Maybe it would have been easier to just normalize all volume size distributions to 1 and then calculate the average values.

> AERONET's volume distribution retrieval is a column-integrated retrieval which is reported per unit area. The scaling was not strictly necessary in this case since we are not comparing AERONET retrievals to lidar retrievals. The purpose of the mean size distributions figure was to show qualitatively the proportion between the coarse and fine modes. In this case the scale factor was chosen as an average mixing layer height estimated from the lidar and in situ measurements. In California the PBL was very shallow and most aerosols were confined within the first kilometer. In Texas the PBL was higher and 3 km was used as the scaling factor.

***Response to technical comments:***

1. Page 5, line 5: Replace 'mum' by µm.

> Fixed.

2. Throughout the manuscript: Please don't put the unit meter in italics.

> Fixed.

3. Page 8, line 22: One web-link is probably sufficient here.

Fixed.

4. Fig. 2: Please use the introduced variables for the scattering and absorption coefficients (y-labels) and avoid abbreviations like Scat450 or Abs532. Instead of 'recalc' it would be better to use an abbreviation like 'retr' (retrieved).

Fixed.

5. Fig. 3: Please use the correct variables for the axis-labelling (see comment above). Units should be next to the numbers and not in the next row.

Fixed.

6. Fig. 2 and 3: To be consistent in the equations, the scattering efficiency should also depend on the dry or ambient particle diameter.

Fixed.

7. The section headings for 5.2 to 5.4, 6.2 are quite long and complicated. They can be shortened to be more concise (e.g. 'Retrieval of dry complex refractive index' or 'Retrieval of the effective growth factor' or 'Optical closure study').

Fixed.

8. Page 13, line 2: Please replace here and throughout text (where possible) $3\beta+2\alpha$ by 'extinction and backscatter coefficients'. This will improve the reading flow since the acronym is very specific for the lidar community and not known to the majority of the readers.

The authors explicitly describe the meaning of 3\beta + 2 \alpha in the introduction.

9. Please replace 'optical particle counters' by 'optical particle size spectrometers'. UHSAS and LAS are not just counting particles like a CPC, they also size them.

Fixed.

10. Fig. 4: Please add that these profiles are for the fine mode fraction only.

Fixed.

11. Fig. 8: Units missing in the statistics text blocks.

Fixed.

12. Fig. 11: Define SZD at the beginning of the caption.

Fixed.

13. Page 17, line 9: κ can range according to Petters and Kreidenweis (2007) up to 1.3 (sea spray).

Fixed.

14. Page 17, line 29: The correct formula for ammonium sulphate is $(NH_4)_2SO_4$

Fixed.

15. Sect. A1: Please harmonize the variable names (i.e. the refractive index is given in different ways).

Fixed.

16. Fig. 1: Replace 'King Air' by 'B-200' as shown in the figure (or vice versa).

Fixed.

17. AOT and AOD are not used in the same way throughout the manuscript. For example, in Fig. 8 it is AERONET AOT and in Fig. 10 it is AOD for the fine fraction. Please harmonize.

Fixed.

---

## Author Comment (AC2) · 1 Sep 2016

**Author comments to Referee #2 (RF2)**

*Response to general comments:*

The authors would like to thank RF2 for his/her thorough review of our manuscript. All his/her comments and suggestions were carefully considered and addressed.

*Response to specific comments:*

1) Page 1, Line 19: This reviewer believes the McFarquhar reference is missing. These references are just examples so should be prefaced with "e.g."
   McFarquhar reference was not missing, but the DOI was missing from the reference. That was fixed. (e.g.) was added to the citation.

2) Page 2, Line 20: This reviewer suggests using "focused on" instead of "in" Page 3, Line 8: This reviewer suggests using "non-representative" instead of "unrepresentative"
   Fixed.

3) Page 3, Lines 10-14: Text makes it sound like only 2 spirals were flown in TCAP. Which sounds incorrect. Please also explain that HSRL-2 was only deployed in phase 1 of TCAP. It would be nice to mention the in situ aircraft by name (DOE G-1) and perhaps use the corresponding reference (Schmid et al., 2014). Definitely must explain why hygroscopic adjustments were not necessary. Also please quantify the level of agreement instead of just saying "good agreement". It might also be worth mentioning comparisons done in TCAP Phase I between HSRL-2 and another remote sensing method (4STAR, Shinozuka et al., 2013a, b).
   Paragraph was modified accordingly. The level of agreement was not explicitly quantified in the original paper by Muller et al, 2014. The sentence was modified to "Agreement within the uncertainties of the two methods was observed between the HSRL-2 retrievals and the in situ measurements". With respect to the hygroscopic corrections, please refer to item 8 of the authors' comments to RF1.

4) Page 4, Line 4: Missing "the" before NASA
   Fixed.

5) Page 5: Line 5: micron symbol got lost
   Fixed.

6) Page 6, Line 25: Please specify that this is the airborne version of UHSAS mounted in the free airstream. Who good is sizing of UHSAS near 1 micron?
   The section title has been modified to "Airborne in situ measurements". The characterization of the UHSAS instrument is presented in Cai et al (2008).

7) Page 8, Line 22: "most" leads me to ask the question "why not all" data are publicly available.

The HSRL-2 microphysical retrievals are still considered a research product under development and therefore have not been uploaded to the public archive. The data are available upon request.

8) Page 9, Line 8: The size range of UHSAS is a subset of the size range the Neph and PSAP sees unless a size cut-off of 1 micron was used which is not mentioned. You need the LAS data for the larger sizes. Have you done a cumulative scattering calculation as in Kassianov et al. to see what percentage of scattering the UHSAS is missing?
That's correct. The revised manuscript discusses that. A cumulative scattering calculation was not done but, by suggestion of Referee #1 (see item 33 of his/her review), we added a comparison of AOT calculated from integrating in situ extinction coefficients to those obtained from AERONET at the spiral sites.

9) Page 11, Line 25: At this point in the manuscript this reviewer was very surprised to learn that only the fine mode had been considered. Finding this out on page 11 is too late and a bit frustrating.
The authors added a sentence in the introduction to note that the comparison of size parameters was limited to the fine mode.

10) Page 12, Line 4: At this point you need to say that the cut-off is likely not sharp at exactly 5 microns. Rather it probably has an S-shape. Some particles larger than the cut-off will still make it through the inlet, whereas some that are smaller than the cut-off won't. So is 5 micron the 50% efficiency point?
At the beginning of Section 4 (In situ instruments) the authors had already mentioned the 50% cutoff efficiency of the P3B inlet at 5 um.

11) Page 12, Line 4: Very late in the manuscript (p.12) to say that supermicron data from LAS were not available for DAQ TX.
We mention that earlier in the revised manuscript.

12) Page 12, Line 11. Good to see this improvement!
It was!

13) Page 12, Line 14. Unclear. Increase of what?
What increases is the backscatter efficiency, which is the ratio between the backscattering cross section to the particle's geometric cross section.

14) Page 20, Line 25. Some strange symbols
Fixed.

15) Figure 2: Should use a different symbol than alpha as this is used already for extinction. In 3 beta +2 alpha.
Fixed in Figures 2 and 3.

---

## Author Comment (AC3) · 1 Sep 2016

**Author comments to Referee #3 (RF3)**

*Response to general comments:*

The authors would like to thank RF3 for his/her thorough review of our manuscript. All his/her comments and suggestions were carefully considered and addressed.

I have one issue with the data analysis. I suspect the real part of dry refractive index is systematically overestimated. That is because of the discrepancy in particle size between the two sets of measurement being compared: The submicron particles that the UHSAS observed are held accountable for the extinction by the particles up to 5 um that the nephelometer and PSAP observed. To reduce the systematic error, one could compare size distribution and extinction for an identical size range. An impactor is commonly used to pass particles under 1 um aerodynamic diameter. Its passing efficiency modeled for geometric diameters (see, for example, paragraph [21] of Howell et al., 2006) allows adjustment of the measured size distribution for the particles behind it. Optimize the dry refractive index for the adjusted dry size distribution and the scattering and absorption measured behind the impactor. The overestimate in refractive index, which I think should be noted in the manuscript, has implications. It invites a systematic bias in the calculated extinction and backscattering except the extinction in the vicinity of the nephelometer and PSAP wavelengths (i.e., 532 nm extinction). So the behind-the-impactor retrieval may help explain the systematic biases shown in Figure 7. This possibility makes it worth trying even if the random error is to be magnified for the smaller coefficients and the uncertainty in the impactor passing efficiency.

The adjustment of the effective refractive index was not performed as suggested by the reviewer, but a discussion has been added regarding the bias that might originate from combining submicron size distributions to scattering and absorption measurements obtained for sub and supermicron particles.

*Response to specific comments:*

1) Page 1. Line 1. Insert "and" after "radii".
   Done.

2) Page 2. Line 18. Replace the slash after dsm with a period.
   Done.

3) Page 3. Line 14. "not necessary". Why? Low ambient RH?
   Please refer to item 8 of the authors' comments to RF1.

4) Page 3. Line 16. Remove "of more than 700 lidar retrievals" because it is said in line 18.
   Done.

5) Page 4. Line 21. Insert ", the latter" after "California".
   Done.

6) Page 5. Line 5. Use the Greek letter instead of mu.
   Done.

7) Page 5. Line 11. The first sentence is unclear. Is it necessary?
   Sentence was removed.

8) Page 6. Line 18. Are these wavelengths correct?
   Wavelengths were corrected.

9) Page 6. Line 29. Replace "proportional" with "related".
   Done.

10) Page 8. Line 13. The vertical resolution of 5 m corresponds to ~1s for typical aircraft vertical speeds. But, while the TSI nephelometer records every second, it does not resolve scattering coefficient for each second. The residence time of particles in the TSI nephelometer is closer to 5s under typical flow rates.
    The in situ measurements are interpolated vertically in order for them to match the HSRL-2 vertical resolution of 15 m for the optical properties, and 75 m or 150 m for the microphysical properties.

11) Page 10. Line 16. Is "approximately" necessary? Also Page 16. Line 8.
    Removed.

12) Page 10. Line 29. Replace "measuremets" with "measurements".
    Done.

13) Page 11. Line 2. Replace "sensitive" with "sensitivity".
    Done.

14) Page 13. Line 15. Replace "seem" with "seems".
    Done.

15) Page 13. Line 28-30. Isn't this because the particles sampled in California were somewhat smaller than those in Texas, as implied in Figure A2? Smaller particles are less prone to inlet loss. Can you show the bias for the 532 nm extinction as a function of the Angstrom exponent? The 532 nm extinction is a good choice here because it should be barely affected by the refractive index bias mentioned above.
    This is discussed in the revised manuscript. We used the ratio between in situ and AERONET AOT and compared to the Angstrom exponent measured with AERONET.

16) Page 16. Line 2. What does "a preliminary assessment . . ." refer to?
    To the preliminary results from sensitive studies that have been performed with simulated data in order to assess the uncertainties of the 3+2 retrievals. We added a reference to Table 1 to make the reference clearer.

17) Page 18. Line 32. Make "I" small.
   Done.

18) Figure 2. Note the particle size range for the dN/dlogD (< 1um) and the measured scattering and absorption (< 5um).
   Done.

19) Figure 3. Make the "O" as large as "H" in the upper right box.
   Done.

20) Figure 4. Indicate that the values refer to the fine-mode only. Perhaps also for Figure 5 and 6.
   Done.

21) Figure 6. Should "q1+1.5xIQR" read "q3+1.5xIQR"? Also, what is the significance of 1.5xIQR? Why is this expression used instead of another set of percentiles like 5% and 95%?
   Fixed. The 1.5xIQR is the standard definition of outliers in statistics.

---

## Author Comment (AC4) · 1 Sep 2016

**Author comments to Referee #4 (RF4)**

***Response to general comments:***

The authors would like to thank RF4 for his/her thorough review of our manuscript. All his/her comments and suggestions were carefully considered and addressed.

***Response to specific comments:***

1) Page 2, line 20-34: Since the main objective of the paper is the evaluation of lidar retrievals with airborne insitu data, some discussion should be included here about what have been done in the past. For example, Granados-Muñoz et al. (2016) compared aerosol volume concentrations retrieved from Lidar measurements with airborne in-situ size-distributions. The paragraph in page 3, line 10 about Müller et al. (2014) could be moved here for consistency.
   Fixed. Reference was included.

2) Page 3, line 14: Why hygroscopic adjustments were not necessary?
   Please refer to item 8 of the authors' comments to RF1.

3) Page 3: sections 2, 3 and 4 headings could be improved. . .
   Fixed.

4) Page 5: aerosol properties are measured in situ onboard the P-3B aircraft. . .
   Fixed.

5) Page 5, line 5: replace "mum" by "µm".
   Fixed.

6) Page 6, line 1: How is the RH< 40% achieved? nafion dryer?
   Correct. A nafion dryer is used to dry the aerosol sample stream.

7) Page 6, line 1: State the actual RH inside the dry nephelometer (mean and std)
   Fixed.

8) Page 6: How well do the two nephelometers compare when measuring at RH<40%?
   In general, the raw data from the two nephelometers usually agree within 2-5/Mm. The nephelometers were corrected for any offset measured when running at the same RH, which was usually less than 2/Mm.

9) Page 6, line 2: I see one major problem here, and it is the calculation of the gamma parameter with only two RHs. Additional discussion about the errors and uncertainties introduced with this approach should be included here. This will be very important for deliquescent aerosols.
   A discussion on that topic has been added to the revised version of the manuscript.

10) Page 6, line 13: The authors refer to aerosol backscattering instead of scattering, right?

Fixed.

11) Page 6, Line 18: Are the PSAP wavelengths correct?

The wavelengths have been corrected.

12) Page 6, line 21: A reference supporting the non-hygroscopic enhancement in the absorption coefficient should be included here.

Fixed. Also, by suggestion of RF1, we included the values of the dry single scattering albedo measured during the campaign.

13) Page 6, line 24-Page 7, line 8: The UHSAS and LAS instruments overlap in the range 0.09-1 μm, have the authors compared the size distributions in the overlapping region? Have these instruments been intercompared with more robust instrument like SMPS?

The UHSAS measurements have been compared to the LAS measurements and good agreement has been observed in the overlap region. However, a systematic comparison among the three instruments has not been performed.

14) Page 7, line 10: only at 550 nm?

Correct. The wet nephelometer only measures at one wavelength: 550 nm.

15) Page 7, lines 14-15: This is a repetition. . .

Fixed.

16) Page 7, lines 23-28: These two paragraphs are confusing; the first one ends with ". . . mdry values are unknown." and the second one starts with "Once mdry is determined. . .". How is mdry determined? It is not said until section 5.2 in page 8! This section is jumpy and messy. . .

Fixed.

17) Page 8, line 30: nephelometer

Fixed.

18) Section 5.2: why the size distribution measured with LAS is not used? The nephelometers are sampling up to 5 μm (inlet cut-off) but the size distribution used is restricted to diameters < 1 μm.

The LAS size distributions were available only during DAQ CA. The results obtained with the LAS data are currently discussed in Section 6.2. Per RF2 suggestion, the authors will make sure to mention the LAS measurements earlier in the paper.

19) Section 5.2: The UHSAS is calibrated using AS (m = 1.53) as stated previously, and then this AS-calibrated size distribution is used to retrieve mdry. Thus, from the reviewer's point of view, the mdry obtained should be seen as an "effective" refractive index able to reproduce your optical

measurements. Therefore, the consistency of the insitu measurements (as stated in page 9, line 10) is not demonstrated.

A note has been added on that and the sentence has been removed.

20) Figure 3: it would be nice to see the ambient RH profile as well.

Fixed

21) Section 5.2 and Figure 3: The authors should keep in mind that the scattering coefficient at ambient conditions is not a measured variable, is a calculated one (and likely with a high uncertainty due to the gamma fit).

That is correct. A discussion on the way gamma is calculated has been added to the manuscript.

22) Page 9, last paragraph: the symbols for the extinction and backscatter coefficient should be introduced earlier. The computed backscatter coefficient, refer to 180° or to integrated hemispheric backscatter?

Fixed. The computed backscatter coefficient refers to 180°.

23) Page 10, 3rd paragraph: repetition

Fixed.

24) Page 10, line 17: why the coarse mode is not studied? The notation in graphs and in the text should make clear that the surface and volume concentrations refer only to the fine mode.

The coarse mode could not be studied in detail due to experimental limitations. We note that in the manuscript.

25) Figure 4: Why these profiles have been shown?

The comparison of profiles has been shown to demonstrate the sensitivity of the lidar retrievals to small changes in the vertical distribution of aerosols.

26) Figure 5: In-situ also refers to fine mode only

Correct. A note has been added.

27) Page 10, line 25: do the authors have any explanation about the different agreement found for the surface and volume concentrations?

At this point we can only speculate. In simulation studies the surface-area usually is the variable which is better reproduced. In our study since the in situ measurements also have uncertainties due to the method used to adjust them for hygroscopicity, it becomes difficult to establish the cause for the difference in agreement that is observed.

28) Page 10, line 27: Any explanations about why the bias was larger in California than in Texas? I would suggest presenting the results combined with the discussion.

That is one of the points which we investigate and report in the manuscript.

29) Section 6.2: It would make more sense to compare the measured in-situ extinction coefficient (scattering + absorption) with the HSRL-2 extinction coefficient, rather than the retrieved insitu extinction coefficient. . .

Due to the strict agreement threshold imposed when retrieving the ambient refractive index, the measured and computed extinction at 532 nm agreed within 1%. Since the only in situ ambient measurement available is the extinction coefficient at 532 nm, we chose to show the computed properties in all plots, instead, for consistency.

30) Figure 8 can be omitted.

Figure 8 has been moved to the Supplemental Material.

31) Section 7: As mentioned before, it would be better to present the results together with their discussion.

The Summary and Discussion have been combined into one section.

32) Page 13, line 23: initially?

That section has been restructured.